# Novel Substituted Thiophenes and Sulf-Polyacetylene Ester from *Echinops ritro* L.

**DOI:** 10.3390/molecules24040805

**Published:** 2019-02-22

**Authors:** Liang-Bo Li, Guang-Da Xiao, Wei Xiang, Xing Yang, Ke-Xin Cao, Rong-Shao Huang

**Affiliations:** Department of Agronomy, Agricultural College of Guangxi University; Nanning 530004, China; llb100@126.com (L.-B.L.); xgd0104@163.com (G.-D.X.); victorxiang@st.gxu.edu.cn (W.X.); Hdyx0000@163.com (X.Y.); 18991697639@163.com (K.-X.C.)

**Keywords:** *Echinops ritro*, thiophenes, sulf-polyacetylene, antimicrobial activities

## Abstract

Three new substituted bithiophenes (**1**–**3**), and one new sulf-polyacetylene ester, ritroyne A (**16**) were isolated from the whole plant of *Echinops ritro* together with twelve known substituted thiophenes. The structures were elucidated on the basis of extensive spectroscopic analysis including 1D and 2D NMR as well as MS. Furthermore, the absolute configuration of ritroyne A (**16**) was established by computational methods. In bioscreening experiments, four compounds (**2**, **4**, **12**, **14**) showed similar antibacterial activity against *Staphylococcus aureus* ATCC 2592 with levofloxacin (8 µg/mL). Five compounds (**2**, **4**, **9**, **12**, **14**) exhibited antibacterial activities against *Escherichia coli* ATCC 25922, with minimum inhibitory concentration (MIC) values of 32–64 µg/mL. Three compounds (**2**, **4**, **12**) exhibited antifungal activities against *Candida albicans* ATCC 2002 with MIC values of 32–64 µg/mL. However, compound **16** did not exhibit antimicrobial activities against three microorganisms.

## 1. Introduction

The genus *Echinops* (Compositae) comprises over 120 species worldwide, widely distributed in Eastern and Southern Europe, tropical and North Africa, and Asia [1]. In Chinese Pharmacopoeias, the dried root of *Echinops latifolius* Tausch and *Echinops grijsii* Hance have been used as a well-known traditional Chinese medicine “Yuzhou Loulu” to relieve heat, expel miasma, and stimulate milk secretion [2]. In China, *Echinops ritro* L. has only been found in Xinjiang province, and is used instead of *E. grijsii* in Uighur Pharmacopeia [3]. Previous phytochemical investigations of *E. ritro* reported the isolation of quinoline alkaloids [4], flavonoids [5], and sesquiterpenes [6], as well as fatty acids [7] and alkanes [8]. In addition, numerous investigations of the genus *Echinops* have resulted in the isolation of thiophenes [9]. Thiophenes from *Echinops* have been proven to possess several activities, such as anti-tumor [10,11], anti-virus [11,12], insecticidal [13], and anti-fungal [14].

Compared with the intensive investigations of the root of this genus, few chemical studies have been conducted on the whole plant of *E. ritro* in recent years [15]. As a part of our ongoing search for bioactive secondary metabolites from Uighur medicinal plants, we herein report the isolation and structure elucidation of fifteen thiophene compounds including three new ones (**1**–**3**) and one new sulf-polyacetylene ester, ritroyne A (**16**), from *E. ritro* as well as their antimicrobial activities.

## 2. Results and Discussion

The 95% ethanolic whole-plant extract of *E. ritro* was subjected to separation using various chromatographic techniques, such as liquid–liquid extraction, silica gel column, octadecylsilyl (ODS) column, Sephadex LH-20 column, and medium-pressure liquid chromatography (MPLC), to obtain three new substituted bithiophenes (**1**–**3**) and one new sulf-polyacetylene ester, ritroyne A (**16**), together with 12 known substituted thiophenes (**4**–**15**) (Figure 1).

### 2.1. Structure Elucidation of the Compounds

Compound **1** was obtained as a primrose-yellow amorphous powder. Its high-resolution electrospray ionization (ESI) mass spectrum displayed a [M + H]^+^ ion peak at *m*/*z* 279.0145, indicating a molecular formula of C_13_H_10_O_3_S_2_. Absorption bands at 3283 cm^−1^ and 2217 cm^−1^ in the IR spectrum were observed, suggesting the existence of hydroxyl and alkyne groups. The ^1^H-NMR spectrum data of **1** (Table 1 and Appendix A), coupled with the ^1^H-^1^H COSY spectrum (Appendix A and Figure 2), showed two sets of low field signals, attributable to four methine protons on two thiophene rings (δ 7.85 and 7.41, each 1H, d, *J* = 4.0 Hz; δ 7.37 and 7.22, each 1H, d, *J* = 3.8 Hz), a triplet (δ 4.56, 1H, t, *J* = 6.5 Hz) and a multiplet (δ 3.67, 2H, m), attributable to a CH(OH)CH_2_OH moiety, and a singlet (δ 9.85, 1H) attributable to aldehyde group proton. The ^13^C-NMR spectrum (Table 1 and Appendix A) showed 13 carbon signals, and characteristic signals including one aldehyde carbon signal, a pair of alkyne carbons signals, eight bithiophene ring signals, and one methoxy group. Careful comparison of the NMR data of **1** with those of a known substituted thiophene compound **7 [16]** revealed that the two compounds were similar, except **1** had an additional aldehyde group that was absent in **7**. Analysis of correlations observed in the HMBC spectrum (Appendix A and Figure 2) allowed the position of the aldehyde group to be elucidated. The H-4 proton signal showed ^1^H-^13^C long-range correlation with C-6, and the H-6 proton signal showed correlations with C-5, indicating that the additional aldehyde group was assignable to C-5. From the above data, the planar structure of **1** was elucidated as 5′-(3,4-dihydroxybut-1-yn-1-yl)-[2,2′-bithiophene]-5-carbaldehyde.

Compound **2** was obtained as a yellowish-white amorphous powder. Accurate mass measurement of an [M + H]^+^ ion peak at *m*/*z* 267.0506 in HR-ESI-MS allowed a molecular formula of C_13_H_14_O_2_S_2_ to be assigned to compound **2**. IR absorption bands at 3277 cm^−1^ for the hydroxyl group and at 1651 cm^−1^ for the carbonyl group were observed. The ^1^H-NMR spectrum of compound **2** (Table 1) displayed two pairs of proton signals (δ 7.12 and 6.71, each 1H, d, *J* = 3.6 Hz; 7.61 and 7.09, each 1H, d, *J* = 4.0 Hz), which indicated the presence of 5,5′-substituted 2,2′-bithiophene moiety. The ^1^H-NMR spectrum of compound **2** also exhibited signals of one methyl protons at δ_H_ 2.50 (s, 3H), one oxygenated methylene protons at δ_H_ 3.75 (t, 2H, *J* = 6.0 Hz), and two methylene protons at δ_H_ 3.04 (t, 2H, *J* = 7.0 Hz) and 1.98–2.05 (m, 2H). The ^13^C-NMR and DEPT spectra of **2** (Table 1) showed 13 carbon signals, attributed to eight bithiophene carbon signals, three methylene signals, one methyl signal, and one carbonyl signal. Careful comparison of the NMR data of **2** with those of compound **13** [17] revealed that compound **2** had one more methyl group than **13**. Analysis of the ^1^H-^1^H COSY and HSQC spectral data of **2** led to the identification of two partial structures: (-CH_2_-CH_2_-CH_2_-OH) and -CH_3_. Based on HMBC correlations (Appendix A and Figure 2) from H-6 to C-4 and C-5, and H-4′ to C-1′′, the methyl and the 4-hydroxybutyryl groups were located at positions C-5 and C-5′, respectively. Thus, on the basis of the above conclusions, the structure of compound **2** was determined to be 4-hydroxy-1-(5’-methyl-[2,2’-bithiophen]-5-yl)butan-1-one.

Compound **3** was obtained as primrose-yellow needle crystals, possessing the molecular formula C_13_H_10_O_4_S_2_ by HR-ESI-MS (*m*/*z* 292.9958 [M – H]^+^). The IR spectrum indicated the presence of hydroxyl group (3426 cm^−1^), carbonyl group (1630 cm^−1^), and alkyne group (2310 cm^−1^). The ^1^H-NMR spectrum of compound **3** (Table 1) showed one oxymethine proton at δ_H_ 4.40 (t, 1H, *J* = 6.0 Hz) and one oxymethylene proton at δ_H_ 3.47 (d, 2H, *J* = 6.0 Hz), attributable to a CH(OH)CH_2_OH moiety, and four overlapped aromatic protons at δ_H_ 7.16–7.19 (m, 4H) suggested that **3** was similar to **1**. The ^13^C-NMR and DEPT spectra (Table 1) were in agreement with the above observations. The only difference between these two compounds was that the aldehyde group in **1** could be oxidized to a carboxylic acid in **3**. This was confirmed by the signal of C-6 at δ_C_ 184.8 in **1** shifted to up-field at δ_C_ 164.3 in **3**. Normally, aldehydes can be converted to acids by strong oxidants. It was difficult to convert compound **1** into **3** under relatively mild conditions during the separation and extraction process. Therefore, compound **3** may be a natural product existing in *E. ritro*. Thus, the planar structure of compound **3** was elucidated as 5′-(3,4-dihydroxybut-1-yn-1-yl)-[2,2′-bithiophene]-5-carboxylic acid.

Ritroyne A (**16**) was obtained as a yellowish powder. The IR spectrum of compound **1** indicated the presence of a hydroxy group (3425 cm^−1^), a carbonyl group (1734 cm^−1^), two C=C bonds (2957, 2932 cm^−1^), and two C≡C bonds (2198, 2126 cm^−1^). Its molecular formula C_21_H_30_O_6_S was determined by HR-ESI-MS experiments (calc. for [M + Na]^+^: 433.1660; found: 433.1664), in combination with ^1^H-NMR and ^13^C-NMR spectral data (Table 2) indicating seven degrees of unsaturation. Analysis of the ^1^H-NMR data and HSQC spectra revealed the occurrence of a pair olefinic protons at δ_H_ 6.38 (dt, 1H, *J* = 16.0, 7.2 Hz) and 5.70 (d, 1H, *J* = 16.0 Hz) connected with a methylene, together with an isolated olefinic proton at δ_H_ 5.91 (s, 1H), six methylenes and three oxygenated methylene protons at δ_H_ 4.34 (d, 2H, *J* = 6.0 Hz), 4.12-4.14 (m, 2H) and 3.69 (t, 2H, *J* = 10.5 Hz), two oxygenated methine protons at δ_H_ 4.36–4.38 (m, 1H) and 3.71–3.74 (m, 1H), and one methyl proton at δ_H_ 0.90 (t, 3H, *J* = 7.6 Hz). The ^13^C-NMR and DEPT spectra showed twenty-one carbon signals, and characteristic signals included four alkyne signals, two oxygenated methylene signals, three oxygenated methylene signals, six methylene signals, one methyl signal, one carbonyl group, and four olefinic signals. The ^1^H-^1^H COSY spectrum (Appendix A and Figure 3) revealed the connectivity of three structural fragments: C-10-C-11-C-12-C-13-C-14-C-15-C-16, C-2-C-3, and C-1′-C-2′-C-3′-C-4′. The connection of C-5 to C-10 via the conjugated diynes was deduced from the observed HMBC correlations (Appendix A and Figure 3) H-12/C-9 and C-10; H-11/C-8 and C-9; H-10/C-6, C-7 and C-8; H-5/C-6 and C-7; and H-17/C-5, C-6, and C-7. Note that HMBC correlations over three bonds were detected, due to the existence of a conjugated system. A similar phenomenon was also observed in the structure of callyberyne A, a C_21_ polyyne from a marine sponge [18]. The position of the S-bond was deduced from the lower field chemical shift of C-3 [δ_C_ 34.7 (t)] and C-4 [δ_C_ 155.8 (s)] compared with the standard value. This was further confirmed by the HMBC correlations H-3/C-4, as well as three bonds correlations H-4/C-3, H-17/C-3, and H-3/C-17 were not detected. The connection of C-2 to C-1′ via the ester group was confirmed by the HMBC correlations H-1′/C-1, H-3/C-1 and 2-OH/C-1. The ∆^(10, 11)^ was assigned as *trans* from the large vicinal coupling constants [^3^*J* (H-10, 11) = 16.0 Hz], and the ROESY (Appendix A and Figure 3) correlation H-12/H-10. The ROESY correlation H-17/H-5 confirmed the ∆^(4, 5)^ was *cis*. Finally, absolute configuration of **16** was determined by our previously constructed matrix method [19,20]. The computed det (D_1_) for C-2 with (*R*) configuration is +8.31 and det (D_2_) for (*R*)-C-14 is −7.49. Four cases, (2*R*,14*R*)-**16**, (2*R*,14*S*)-**16**, (2*S*,14*R*)-**16**, and (2*S*,14*S*)-**16** were investigated, sum of det (D_i_) (i = 1 and 2) values for the four cases were computed to be +6.45, +1.39, −1.39, and −6.45, respectively. The recorded optical rotation value [α]_D_ was +32.1, thus, the *k_0_* values ([α]_D_/det (D)) for the four cases were 5.0, 23.1, −23.1, and −5.0, respectively. Due to the requirement that *k_0_* values should be positive for tertiary alcohols or amines, two cases with negative *k_0_* values ((2*S*,14*R*)-**16** and (2*S*,14*S*))-**16** could be excluded. Considering that the *k_0_* values for the tertiary alcohols are generally from 0.5–6.0 in different chiral alcohols, only the case where *k_0_* is 5.0 matched all conditions. The predicted absolute configuration for **16** is (2*R*,14*R*).

The twelve known substituted thiophene compounds isolated from *E. ritro* L. were identified as arctinol b (**4**) [21], 4-(5-(penta-1,3-diyn-1-yl)thiophen-2-yl)but-3-yne-1,2-diol (**5**) [22], [2,2′-bithiophene]-5-carboxylic acid (**6**) [23], 4-([2,2′-bithiophen]-5-yl) but-3-yne-1,2-diol (**7**) [24], junipic acid (**8**) [23], arctinal (**9**) [24], 4-(5′-methyl-[2,2′-bithiophen]-5-yl)but-3-yn-1-ol (**10**) [25], ethanone (**11**) [26], 4-([2,2′-bithiophen]-5-yl)but-3-yn-1-ol (**12**) [17], 1-([2,2′-bithiophen]-5-yl)-4 -hydroxybutan-1-one (**13**) [17], arctinol A (**14**) [25], and arctic acid (**15**) [27], respectively, by comparison of their spectral data (^1^H, ^13^C-NMR and MS) with those reported in the literature.

### 2.2. In Vitro Antimicrobial Activity

All the sulfur-containing compounds isolated from *E. ritro* were tested for their potential antimicrobial effects on *S. aureus* ATCC 2592, *E. coli* ATCC 25922, and *C. albicans* ATCC2002. The antimicrobial abilities of these compounds isolated from *E. ritro* are shown in Table 3. Among the 16 compounds tested, **2**, **4**, **9**, **12**, and **14** showed antibacterial activities against *S. aureus* ATCC 2592 and *E. coli* ATCC 25922. Compounds **2**, **4**, and **12** exhibited antifungal activities against *C. albicans* ATCC 2002. Thiophenes from *Echinops* have been reported to possess many biological activities, including insecticidal [28] and fungicidal [29]; however, antibacterial activities of thiophenes have rarely been reported. Compounds **5** and **12** which were isolated from *E. ritro* have been reported to exhibit significant antifungal activity against *Colletotrichum* species and *Fusarium* [14], but **5** showed no antimicrobial activity according to our research. Thus, the findings obtained in this study are consistent with the previous antimicrobial studies revealing the potent antibacterial activities of thiophene-type compounds, which deserve further studies on the way to discover new antibacterial agents.

## 3. Experimental Section

### 3.1. General Procedures

Infrared radiation (IR) spectra were measured on a Bio-Rad FTS-135 FTIR spectrometer (Hercules, CA, USA) on KBr pellets. Optical rotations were measured using a JASCO P-1020 automatic digital polarimeter (Easton, PA, USA). NMR spectra were recorded on a Bruker AM-400 and DRX-500 spectrometer (Karlsruhe, Germany) using standard Bruker pulse programs. Chemical shifts were shown as δ-values with reference to tetramethylsilane (TMS) as an internal standard. The ESI-MS and HR-ESI-MS were recorded on an API Qstar Pulsa LC/TOF spectrometer (Foster City, CA, USA). Sephadex LH-20 (Pharmacia Fine Chemical Co. Ltd. Uppsala, Sweden), silica gel (Qingdao Ocean Chemical Co., Ltd., Qingdao, China), and ODS (40–63 µm, Merck, Darmstadt, Germany) were used for column chromatography (CC). Thin-layer chromatography (TLC) was carried out on preparative Silica gel 60 F254 and RP-18 F254 plates (Merck, Darmstadt, Germany), and spots were visualized by spraying the plates with 10% H_2_SO_4_/ethanol, and heating them at 105 °C. A Buchi Sepacore System (Fällanden, Switzerland) was used for medium-pressure liquid chromatographic (MPLC) separations.

### 3.2. Plant Material

Air-dried whole plants of *Echinops ritro* L. were obtained from Qinghe County, Xinjiang Province, China, in August 2007, and were identified by Prof. Rongshao Huang of Agricultural College of Guangxi University in Nanning. A voucher specimen (No. 20070810) was maintained in the lab of Agricultural College of Guangxi University in Nanning (530004), China.

### 3.3. Extraction and Isolation

The air-dried whole plants of *E. ritro* (30 kg) were extracted with 95% EtOH (150 L × 3) and refluxed. Evaporation of the organic solvent under reduced pressure at 55 °C yielded a crude extract (5500.0 g). The concentrated brown syrup was resuspended in water and partitioned with ethyl acetate (15.0 L × 3) and water-saturated n-butanol (10.0 L × 3) gradually to afford 500.0 g and 450.0 g of dried organic extracts, respectively.

The ethyl acetate fraction (500.0 g) was absorbed on 750 g silica gel and fractionated over a silica gel (4 kg, 200–300 mesh, 10 × 150 cm) column by eluting gradually with CHCl_3_:MeOH (100:1 (30 L), 50:1 (30 L), 20:1 (30 L), 10:1 (20 L), 5:1 (20 L), 2:1 (20 L)) to yield six main fractions, namely A–F. Fraction A (13.0 g) was subjected to medium-pressure liquid chromatography (SiO_2_, 1300 g) eluting with stepwise petroleum ether–acetone gradient (0% to 50% acetone) and purified by Sephadex LH-20 (20 g) CC (MeOH:CHCl_3_ = 1:1, 200 mL) to afford **1** (6 mg), **4** (120 mg), and **9** (160 mg). In a similar way, compounds **5** (14 mg) and **7** (23 mg) were obtained from fraction B (30 g). Fraction E (55.0 g) was subjected to preparative medium-pressure liquid chromatography (C18-MPLC, 1600 g) eluting with stepwise H_2_O–MeOH gradient (10% to 90% MeOH) to obtain fractions E_1–3_. Purification of E_2_ (1800 mg) by medium-pressure liquid chromatography (SiO_2_, 200 g) eluting with stepwise petroleum ether–acetone gradient (5% to 60% acetone) yielded compounds **8** (16 mg), **11** (11 mg), **12** (25 mg), and **13** (21 mg), as well as an additional fraction containing impure **2**. Compound **2** (8 mg) was also purified from the latter fraction (20 mg) by Sephadex LH-20 (20 g) CC (MeOH, 200 mL). Fraction E_3_ (2.0 g) was subjected to medium-pressure liquid chromatography (SiO_2_, 240 g) eluting with stepwise petroleum ether–acetone gradient (0% to 40% acetone) to give compounds **6** (51 mg), **14** (23 mg), and **15** (10 mg). Fraction F (3.5 g) was fractionated over a Sephadex LH (250 g) column (MeOH, 1500 mL) to obtain three main subfractions, fractions F_1–3_. Rechromatography of fraction F_2_ (500 mg) by MPLC (SiO_2_, 60 g) eluting with stepwise petroleum ether–acetone gradient (10% to 60% acetone) afforded compounds **3** (60 mg), **10** (16 mg), and **16** (40 mg). The structures of new compounds **1**–**3** and **16** are shown in Figure 1.

### 3.4. Compound Characterization Data

5′-(3,4-Dihydroxybut-1-yn-1-yl)-[2,2′-bithiophene]-5-carbaldehyde (**1**). Primrose-yellow amorphous powder; IR (KBr, cm^−1^) *v*_max_ 3282 (OH), 3087, 2963, 2927, 2217, 1650, 1455, 1436, 1233, 1086, 1057; ^1^H-NMR (500 MHz, methanol-*d_4_*) and ^13^C-NMR (125 MHz, methanol-*d_4_*) spectroscopic data, see Table 1; EI-MS (*m*/*z*): 278 [M]^+^, 260 [M − H_2_O]^−^, 247, 231, 218, 190, 171, 158, 145, 95, 69; HR-ESI-MS (*m*/*z*): 279.0145 [M + H]^+^, Calc. for [C_13_H_11_O_3_S_2_ + H]^+^ 279.0149.

4-Hydroxy-1-(5′-methyl-[2,2′-bithiophen]-5-yl)butan-1-one (**2**). Yellowish-white amorphous powder; IR (KBr, cm^−1^) *v*_max_ 3277 (OH), 3065, 2953, 2863, 1651, 1509, 1453, 1207, 1052, 1014, 807; ^1^H-NMR (500 MHz, CDCl_3_) and ^13^C-NMR (125 MHz, CDCl_3_) spectroscopic data, see Table 1; EI-MS (*m*/*z*): 266 [M]^+^, 248 [M − H_2_O]^−^, 222, 207, 179, 149, 135, 91, 69; HR-ESI-MS (*m*/*z*): 267.0506 [M + H]^+^, calcd for [C_13_H_14_O_2_S_2_+H]^+^ 267.0513.

5′-(3,4-Dihydroxybut-1-yn-1-yl)-[2,2′-bithiophene]-5-carboxylic acid (**3**). Primrose-yellow needles crystals, mp.: 216–220 °C; IR (KBr, cm^−1^) *v*_max_ 3426 (OH), 2311, 1630, 1552, 1434, 1379, 1320, 1149, 1105, 766; ^1^H-NMR (500 MHz, CD_3_OD) and ^13^C-NMR (125 MHz, CD_3_OD) spectroscopic data, see Table 1; ESI-MS (*m*/*z*): 293 [M − H]^+^, 277, 249; HR-ESI-MS (*m*/*z*): 292.9958 [M − H]^+^, calcd for [C_13_H_9_O_4_S_2_−H]^+^ 292.9942.

Ritroyne A (**16**). yellow solid; [α = +32.1 (c 0.29, methanol); IR (KBr, cm^−1^) *v*_max_ 3452 (OH), 2957, 2932, 2872, 2198, 2126, 1734, 1627, 1457, 1093; ^1^H-NMR (400 MHz, acetone-*d*_6_) and ^13^C-NMR (100 MHz, acetone-*d*_6_) spectroscopic data, see Table 2; EI-MS (*m*/*z*): 410, 392, 365, 347, 335, 322, 310, 294, 219, 128, 115; HR-ESI-MS (*m*/*z*): 433.1664 [M + Na]^+^, calcd for [C_21_H_30_O_6_S+Na]^+^ 433.1660.

### 3.5. Assay for Antimicrobial Activity

The assays were performed in a similar manner as previously reported [30] using *S. aureus* ATCC2592, *E. coli* ATCC25922, and *C. albicans* ATCC2002. EBS medium [31] for bacteria and MYC-medium (1.0% phytone peptone; 1.0% glucose; 50 mM HEPES (11.9 g/L) pH 7) for fungi was used, respectively. Briefly, 100 µL of bacteria suspensions were seeded into each well of 96-well cell culture plates containing 10^6^ CFU/mL, exposed to the test compounds and positive control (levofloxacin) at concentrations of 516, 256, 128, 64, 32, 16, 8, 4, 2, 1 µg/mL at 37 °C for 24 h. All experiments were repeated in triplicate and average MICs are reported in Table 3.

## 4. Conclusions

Three new substituted bithiophenes (**1**–**3**), and one new sulf-polyacetylene ester, ritroyne A (**16**), together with twelve known substituted thiophenes (**4**–**15**), were isolated from the 95% ethanolic extract of the whole plant of *E. ritro*. The polyacetylenes are commonly considered as precursors of thiophenes occurring in the Compositae, but sulf-polyacetylenes were rarely reported from previous phytochemical investigations [32]. To the best of our knowledge, this is also the first report on the isolation of sulf-polyacetylene fromthe *Echinops* genus. In addition to the phytochemical work, the in vitro biological activities of isolates were evaluated against three microorganisms (*S. aureus*, *E. coli*, and *C. albicans*) using a microdilution method. Compounds **2**, **4**, **9**, **12**, and **14** exhibited antibacterial activities against *S. aureus* ATCC 2592 and *E. coli* ATCC 25922. Compounds **2**, **4**, and **12** exhibited antifungal activities against *C. albicans* ATCC 2002.

## Figures and Tables

**Figure 1 molecules-24-00805-f001:**
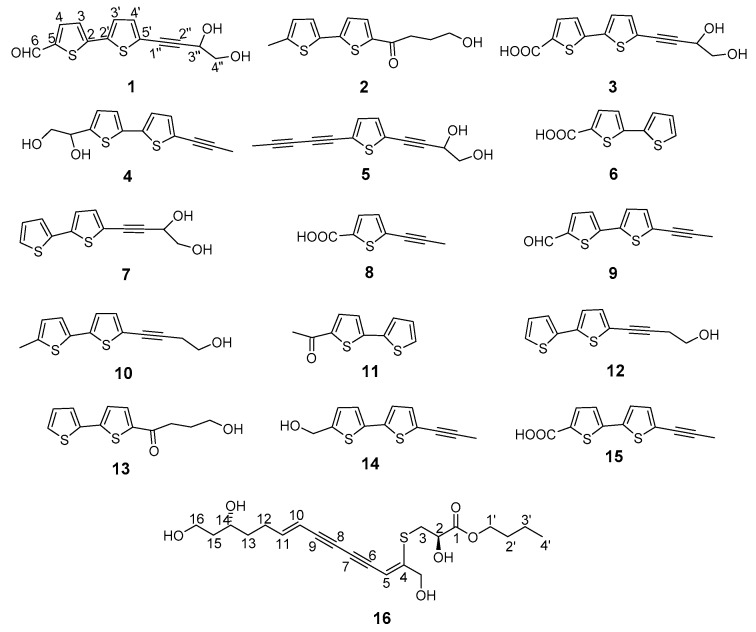
Chemical structures of compounds **1**–**16** from *Echinops ritro* L.

**Figure 2 molecules-24-00805-f002:**
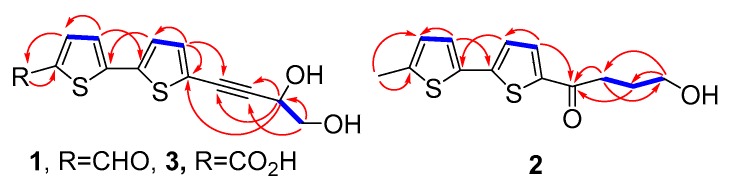
^1^H, ^1^H-COSY (
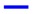
) and key correlations in HMBC spectra (H
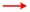
C) of compounds **1**–**3**.

**Figure 3 molecules-24-00805-f003:**
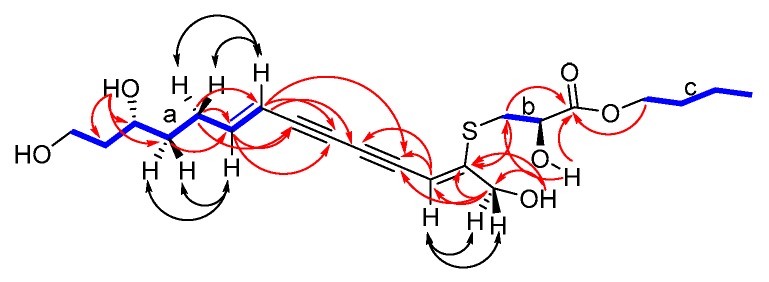
^1^H, ^1^H-COSY (
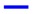
), key HMBC (H
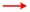
C), and key ROESY (
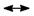
) correlations of **16**.

**Table 1 molecules-24-00805-t001:** ^1^H-NMR (500 MHz) and ^13^C-NMR (125 MHz) data of **1**–**3** (^a^ is recorded in methanol-*d*_4_; ^b^ is recorded in CDCl_3_) (δ in ppm, *J* in Hz).

	**1** ^a^		**2** ^b^		**3** ^a^	
	δ_C_	δ_H_ mult (*J*)	δ_C_	δ_H_ mult (*J*)	δ_C_	δ_H_ mult (*J*)
2	146.9		134.0 s		138.6 s	
3	126.4	7.41 d (4.0)	125.7 d	7.12 d (3.6)	123.7 d	7.16–7.19 m
4	139.6	7.85 d (4.0)	126.5 d	6.71 d (3.6)	133.4 d	7.16–7.19 m
5	143.6		141.3 s		135.5 s	
6	184.8	9.85 s	15.4 q	2.50 s	164.3 s	
2′	138.3		146.3 s		128.5.3 s	
3′	127.3	7.37 d (3.8)	123.4 d	7.09 d (4.0)	124.6 d	7.16–7.19 m
4′	134.7	7.22 d (3.8)	133.0 d	7.61 d (4.0)		7.16–7.19 m
5′	125.4		141.7 s		120.4 s	
1′′	78.2		193.0 s		76.7 s	
2′′	95.8		35.5 t	3.04 t (7.0)	96.1 s	
3′′	64.7	4.56 t (6.5)	27.2 t	1.98–2.05 m	63.1 d	4.40 t (6.0)
4′′	67.0	3.67 m	62.3 t	3.75 t (6.0)	65.5 t	3.47 d (6.0)

**Table 2 molecules-24-00805-t002:** ^1^H- (400 MHz) and ^13^C-NMR (100 MHz) data of **16** in acetone-*d*_6_ (δ in ppm, *J* in Hz).

	**16**	δ_C_		δ_H_	δ_C_
	δ_H_				
1		179.2	11	6.38 dt (16, 7.2)	149.8
2	4.36–4.38 m	71.5	12	2.26–2.29 m	29.6
3	3.39 dd (11.5, 4.4)	34.7	13	1.50–1.54 m	37.3
	3.25 dd (11.5, 4.4)				
4		155.8	14	3.71–3.74 m	69.5
5	5.91 s	103.7	15	1.59–1.61 m	40.4
6		81.8	16	3.69 t (10.5)	60.5
7		78.4	17	4.34 d (6.0)	64.6
8		73.2	1′	4.12–4.14 m	65.5
9		83.5	2′	1.62–1.64 m	31.3
10	5.69 d (16)	109.2	3′	1.36–1.40 m	19.7
2-OH	4.77 d (6.0)		4′	0.90 t (7.6)	13.9
17-OH	4.59 t (6.0)		14-OH	3.85 d (4.8)	

**Table 3 molecules-24-00805-t003:** Antimicrobial activities (MIC) of the compounds (**1**–**4**, **9**, **12**, **14**, **16**).

MIC (µg/mL)			
Compounds	*S. aureus* ATCC 2592	*E. coli* ATCC 25922	*C. albicans* ATCC 2002
**1**	128	256	256
**2**	8	32	32
**3**	256	>512	>512
**4**	8	64	64
**9**	32	64	>512
**12**	8	64	64
**14**	8	64	>512
**16**	>512	>512	>512
levofloxacin	8	16	64

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
