# Peer review of "Novel Substituted Thiophenes and Sulf-Polyacetylene Ester from Echinops ritro L."

_molecules, 2019, doi:10.3390/molecules24040805_

Round 1

Reviewer 1 Report

The authors have presented their investigation on three new substituted bithiophenes (1-3), and onenew sulf-polyacetylene ester, ritroyne A (16) which were isolated from the Echinops ritro together with twelve known substituted thiophenes. The structures were elucidated based on extensive spectroscopic analysis including 1D and 2D NMR as well asHR-MS.  In addition, the isolated compounds were investigated for their antimicrobial activity.  Overall study concludes, E. ritrois a potential plant source for further research targeting bacteria and fungi diseases.

However, some issues need to be clarified before considering for the publication.

The authors should explain the selection of the plant, whole part and hypothesis of the present work.Authors have tested antibacterial activity direct MIC investigation, need to explain or include any data like disc diffusion, which can reveal activity with low quantity and very promising activity testing. The second step may be MIC. Table.3 showing more than 200 micro gram for some compounds in that case what about the solubility/ turbidity and how much yield was achieved during isolation. The methods part stating that MIC were repeated in triplicate and average values are tabulated need to correct values in table.3. The manuscript abstract, Introduction, methods part showing more than 40% plagiarism.

The manuscript need major revisions and should be stringent manuscript, which has been carefully proof read by the authors before re-submission.  However, the overall writing of the manuscript need English correction

Author Response

The manuscript has been revised the manuscript according to the reviewers' comments, and corrected by English editing service provided by MDPI

Reviewer 2 Report

The manuscript entitled "Substituted thiophenes and one sulf-polyacetylene ester compounds from Echinops ritro L. and their antimicrobial activities" reports an interesting study of bithiophenes and one sulf-polyacetylene ester isolated from Echinops ritro. The manuscript can be accepted. However, there are few point that must be improved.

1 - The values of activities of all compounds must be expressed in molar concentrations. 

2 - There are no insights into the structures of the compounds and its activities. It is extremely suitable to add some SAR studies based on fo the knowledge in the literature or using some in silico methods. Therefore add a section that can increase the importance of the information of the obtained results related to the biological activities of the compounds.

Author Response

(The authors gave the same response as above.)

Reviewer 3 Report

Dear authors

Please, read my review report at the attached file.

Author Response

(The authors gave the same response as above.)

Reviewer 4 Report

Comments and Suggestions for Authors

I read with interest the manuscript entitled: “Substituted Thiophenes and One Sulf-polyacetyleneEster Compoundsfrom Echinops ritro L. and Their Antimicrobial Activities. The manuscript provides a modest advance in organic synthesis. There are many typographical errors, to see attached file named molecules -399990-per-review-v1-revised.

The content of above mentioned manuscript is well written.

However, it requires major corrections and additions.

For example:

The authors indicate that these compounds have antimicrobial activities. However, they tested only 1 strain of each microorganism S aureus ATCC 2592, E. coli ATCC 25922 and C. albicans ATCC 2002. Statistically, these results do not have a weighted average, hence no adequate representation to be considered antibacterial. In this case, it would take at least about 5-10 strains of each to suggest such results. Even these strains are marketed as ATCC microorganisms. Although they do provide some antibacterial indications, but it is not a realistic indication. Biological tests would have to have been done with strains isolated from animals with manifestations of bacterial infections to validate such activities. Consequently, I have not validated the mentioned antibacterial activities of the compounds claimed by the authors. Therefore, the title of the manuscript is not consistent with the study.

The MIC value of 256 ug/mL for some compounds is too high to suggest antibacterial activity.

On the other hand, I have my doubts that the new substituted bithiophenes (2), is unheard of, as it may be a "satellite" product derived from the alkyne 10 through its hydration in some of the extraction or purification steps.

The absolute stereochemistry of compound 16 could have been elucidated by X-ray diffraction, as it is a solid.

Reference 9 is repeated ... with 3.!

Overall, I believe that this manuscript could benefit of a major revision before publication in Molecules.

Author Response

(The authors gave the same response as above.)

Round 2

Reviewer 1 Report

The authors have satisfactorily responded to all my questions and made the necessary changes to the manuscript.

Author Response

Thank you for your comments concerning our manuscript. The modified version of the manuscript will be further modified by English editing provided by MDPI through the editor.

We look forward to hearing from you regarding our submission. We would be glad to respond to any further questions and comments that you have raised.

Reviewer 3 Report

Dear authors

The manuscript by Li et al. (article ID molecules- 399990) was modified but wasn’t improved.

In fact, the authors change several points of manuscript, they eliminate some weaknesses but do not correct others and introduce new weaknesses.

For example, they change the introduction but the end result is that an introduction does not provide enough information and does not include all relevant references. The introduction does not indicate the diseases treated in traditional medicine with the species under study, does not present previous phytochemical studies in the species in studies, and makes statements based on the literature without citing the respective references (e.g. “Compared with the intensive investigations of the root of this genus, few chemical studies have been conducted on the whole plant of E. ritro.”

Other significant example: To support the authors sentence “…a singlet (δ 9.85, 1 H) attributable to aldehyde group proton …” and the data showed in table 1, the HSQC NMR spectra of compound 1 should include the correlation between the signal at δH 9.85 (s, 1 H) and the 13C signal at δC 184.8. This correlation is essential to confirm the aldehyde group. The reviewer's comment was that a fuller, higher resolution spectrum was provided and the authors again have the same spectrum where the correlation is not shown.

Why do the authors provide spectra of compound 4 in the supplementary files? They should present the spectra of compound 16. If the spectra are from compound 16, please correct the figure caption!

The authors use the phrase “The 13C NMR and DEPT spectra of 1 (table 1).”, and “The 13C NMR and DEPT spectra of 2 (Table 1)…”. However, as referred on the previous reviewer report, table 1 doesn’t show any DEPT data. The authors should change the sentences. For example, for compound 2 it could be “The 13C NMR and DEPT spectra of 2 (table 1 and Figure S11)…”. However, for compound 1 the authors don’t provide DEPT data even in supplementary files.

Table 3 has ml instead mL.

The structure/activity relationship of the compounds should be discussed in this paper, once the structures and activities in this work are presented here. Indicate as future work is not acceptable if you have the data here. The discussion is poor and should be enriched by this discussion.

The expression “potent” in the conclusion is too much…

In addition, the authors present an unclean version, difficult to read which makes the reviewers work very difficult.

Author Response

Thank you for your comments concerning our manuscript. We have studied the comments carefully and have made correction which we hope meet with your approval.

Reviewer 4 Report

The manuscript describes Novel Thiophenes and Sulf-polyacetylene Ester from Echinops ritro L.The manuscript provides a modest advance but is worthy of publication. I recommend the publication of this manuscript with minor revisions as indicated in the attached file.

For example:

1.      An absortion band should be: Absortion bands

2.      ug/ml should be: ug/mL

3.      v/v should be deleted and standardizing in manuscript thoroughly.

4.      25mg should be: 25 mg

5.      And other

Author Response

Thank you for your comments concerning our manuscript. We have studied the comments carefully and have made correction which we hope meet with your approval. The modified version of the manuscript will be further modified by English editing provided by MDPI through the editor. We look forward to hearing from you regarding our submission. We would be glad to respond to any further questions and comments that you have raised.